# Autonomic and Neuroendocrine Reactivity to VR Game Exposure in Children and Adolescents with Obesity: A Factor Analytic Approach to Physiological Reactivity and Eating Behavior

**DOI:** 10.3390/nu17152492

**Published:** 2025-07-30

**Authors:** Cristiana Amalia Onita, Daniela-Viorelia Matei, Laura-Mihaela Trandafir, Diana Petrescu-Miron, Calin Corciova, Robert Fuior, Lorena-Mihaela Manole, Bogdan-Mircea Mihai, Cristina-Gena Dascalu, Monica Tarcea, Stéphane Bouchard, Veronica Mocanu

**Affiliations:** 1Center for Obesity BioBehavioral Experimental Research, Department of Morpho-Functional Sciences II (Pathophysiology), “Grigore T. Popa” University of Medicine and Pharmacy, 700115 Iaşi, Romania; 2Faculty of Medical Bioengineering, “Grigore T. Popa” University of Medicine and Pharmacy, 700588 Iaşi, Romania; calin.corciova@umfiasi.ro (C.C.);; 3Center for Diagnosis, Counseling, and Monitoring of Obese children, Department of Mother and Child (Pediatrics), “Grigore T. Popa” University of Medicine and Pharmacy, 700115 Iaşi, Romania; 4Unit of Diabetes, Nutrition and Metabolic Diseases and Internal Medicine, “Grigore T. Popa” University of Medicine and Pharmacy, 700115 Iaşi, Romania; bogdan.mihai@umfiasi.ro; 5Emergency Clinical Hospital “Sf. Spiridon”, 700111 Iaşi, Romania; 6Department of Medical Informatics and Biostatistics, Faculty of Medicine, “Grigore T. Popa” University of Medicine and Pharmacy of Iaşi, Romania, eron 67, 700050 Iaşi, Romania; cristina.dascalu@umfiasi.ro; 7Department of Community Nutrition & Food Safety, University of Medicine, Pharmacy, Science and Technology, 540139 Târgu Mureș, Romania; 8Département de Psychologie et de Psychoéducation, Université du Québec en Outaouais, Gatineau, QC J8X 3X7, Canada

**Keywords:** childhood obesity, virtual reality, immersive VR game, heart rate variability, salivary cortisol, salivary alpha-amylase, eating behavior, three-factor eating questionnaire (TFEQ)

## Abstract

**Background/Objectives**: The aim was to identify patterns of autonomic and neuroendocrine reactivity to an immersive virtual reality (VR) social-emotional stressor and explore their associations with perceived stress and eating behavior. **Methods**: This one-group pretest–posttest study included 30 children and adolescents with obesity (15 boys and 15 girls), aged 8 to 17 years. The VR protocol consisted of two consecutive phases: a 5 min relaxation phase using the Forest application and a 5 min stimulation phase using a cognitively engaging VR game designed to elicit social-emotional stress. Physiological responses were measured using heart rate variability (HRV) indices and salivary stress biomarkers, including cortisol and alpha amylase. Subjective stress and eating responses were assessed via visual analogue scales (VAS) administered immediately post-exposure. The Three-Factor Eating Questionnaire (TFEQ-R21C) was used to evaluate cognitive restraint (CR), uncontrolled eating (UE), and emotional eating (EE). **Results**: The cortisol reactivity was blunted and may reflect both the attenuated HPA axis responsiveness characteristic of pediatric obesity and the moderate psychological challenge of the VR stressor used in this study. Two distinct autonomic response patterns were identified via exploratory factor analysis: (1) parasympathetic reactivity, associated with increased RMSSD and SDNN and decreased LF/HF, and (2) sympathetic activation, associated with increased heart rate and alpha-amylase levels and reduced RR intervals. Parasympathetic reactivity was correlated with lower perceived stress and anxiety, but also paradoxically with higher uncontrolled eating (UE). In contrast, sympathetic activation was associated with greater cognitive restraint (CR) and higher anxiety ratings. **Conclusions**: This study demonstrates that immersive VR game exposure elicits measurable autonomic and subjective stress responses in children and adolescents with obesity, and that individual differences in physiological reactivity are relevantly associated with eating behavior traits. The findings suggest that parasympathetic and sympathetic profiles may represent distinct behavioral patterns with implications for targeted intervention.

## 1. Introduction

The global rise in childhood obesity presents significant long-term health risks. Despite efforts to counter this trend, systematic reviews of various interventions have yielded mixed results. Chronic stress is a key factor, often causing irregular eating habits that contribute to weight gain and obesity. Understanding the neural and hormonal pathways that control stress-induced eating behaviors is essential for developing effective therapeutic targets to prevent and treat obesity [1].

Physiological changes occur in response to environmental changes or threats, either real or imagined [2], helping the body adapt to and overcome stressors. In response to a stressor, the brain activates the autonomic nervous system (ANS) (which results in increases in salivary alpha-amylase, blood pressure, and heart rate) and the hypothalamic–pituitary–adrenal (HPA) axis (which results in increases in salivary cortisol) [3]. The autonomic nervous system (ANS) can respond within seconds to minutes and is indicated through heart rate variability metrics [3,4]. ECGs can analyze heart rate (HR) and heart rate variability (HRV) and indicate the level of stress after observing changes in the autonomic nervous system (ANS) [5]. Neuroimaging studies suggested that HRV may be linked to cortical regions (e.g., the ventromedial prefrontal cortex) involved in stressful situation appraisal [5]. HRV tends to appear low under stress and high under relaxation because HRV indicates that the higher the variation in the normal range or the more complex the pattern is, the higher the adaptability of the ANS to the stressors is [6]. The ANS can also be influenced by neurotransmitters on the timescale of minutes and indicated through alpha-amylase and secretory Ig A concentrations in saliva. In the order of tens of minutes, the hypothalamic–pituitary–adrenal (HPA) axis can release hormones like cortisol that affect target organs [7,8].

Eating behaviors related to stress vary based on the type and duration of stressors, the type of food, and individual susceptibility factors [9]. Over the past decades, numerous studies have measured autonomic nervous system (ANS) and hypothalamic–pituitary–adrenal (HPA) responses to various stressors (e.g., postural challenges, video game challenges, exposure to cold, mental arithmetic challenges). These studies hypothesize that hyperactivity of stress pathways stimulates an increase in energy intake and shifts food choices toward unhealthy items or “comfort foods” [10]. However, other research suggests there is no association between increased adiposity and stress pathway activation, or that increased adiposity is associated with lower stress pathway activation in response to similar stressors [3,11].

Immersive virtual reality (VR) experiences can simulate stressful situations, eliciting physiological stress responses similar to those experienced in real-life situations [12]. VR content creators can deliberately design experiences to evoke stress or anxiety for educational, therapeutic, or entertainment purposes. A study by Aliyari et al. [13] showed that exposure to virtual games can induce stress by activating cortical regions implicated in emotions and cognition. Acute stress in VR can trigger an acute sympathetic-mediated stress response (affecting pulse, blood pressure, skin conductance, and salivary amylase). These responses mimic the body’s natural reactions to stressful situations and can occur even though users are aware that they are in a virtual environment [14,15].

This study investigated stress reactivity in children and adolescents with obesity by assessing their physiological and subjective responses to an immersive virtual reality (VR) social-emotional stressor in the following manner: 1. Evaluating the impact of a VR game-based experimental model on physiological and psychological stress markers, using heart rate variability (HRV), salivary stress markers, and subjective stress reactivity scales; 2. Identifying latent patterns of autonomic and neuroendocrine reactivity using exploratory factor analysis and examining their associations with perceived stress and eating patterns, using the Three-Factor Eating Questionnaire (TFEQ) to assess individual differences in cognitive restraint, uncontrolled eating, and visual analogue scales (VAS) to measure subjective appetite.

In the present study, we employed a VR game-based experimental model designed to act as a mild, non-threatening stimulation intended to elicit physiological reactivity, rather than a classical psychosocial stressor such as the Trier Social Stress Test (TSST). This paradigm aimed to assess autonomic and neuroendocrine responsiveness without inducing high emotional distress.

To date, few studies [16,17,18] have directly examined the relationship between VR game-based stimulation, autonomic nervous system responses, particularly heart rate variability (HRV), and eating behavior in children with obesity.

## 2. Methods

### 2.1. Study Participants

Thirty children aged between 8 and 17 years (15 boys and 15 girls) were recruited from the “Santa Maria” Emergency Clinical Hospital for Children, Iaşi (Romania). The median age was 10 years, with an interquartile range (IQR) of 9–14 years. The inclusion criteria for participants were defined as follows: (i) age range between 8 and 18 years old; (ii) meeting WHO criteria for the diagnosis of obesity in adolescents; (iii) absence of epilepsy; (iv) absence of eating disorders, psychiatric disorders, chronic conditions, or any drug treatment. The age range was selected to ensure participants had sufficient cognitive and emotional maturity to understand and engage meaningfully with the virtual reality (VR) game-based experimental task. Inclusion criteria were as follows: the 30 participants had a BMI-for-age value > +1 SD. Data were gathered between January and March 2024.

### 2.2. The VR-Exposure

The virtual environments were developed with the software Unity3D (Version 6.0) (www.unity3d.com, Unity Software Inc., San Francisco, CA, USA). They were displayed using an Oculus Quest 2 headset (Oculus, Irvine, CA, USA) connected to an Intel^®^ Core (TM) i7-10700 computer (Intel, Santa Clara, CA, USA) with a 2.9 GHz core processor unit and 32.0 Gb of RAM and an Nvidia Geforce RTX 3070 (Nvidia, Santa Clara, CA, USA) graphics card (Figure 1). The lab room had only a desk, a chair for the research assistant, and an armless chair for the participant (Figure 2).

Forest app. A computer-generated forest was created using Unity VR, offering a fully immersive environment with real-time interactive graphics. This virtual forest was designed using high-resolution images and featured various sound effects, such as birds chirping and flowing water. Studies have indicated that exposure to natural environments can have a calming effect.

Game app. The second virtual environment is a skill-based game played in immersion. The game involved rapidly selecting the correct shape to let objects pass objects of various shapes that are coming toward the user on a rail. The objects’ shapes and speeds dynamically change while moving on the rail, and the number of rails increases over time. Background music, a scoring system displaying both the high score and the current score, and the competitive nature of the game were all incorporated to potentially elevate participants’ stress levels. The VR game used a non-threatening format; failure to hit the correct shape was indicated by a subtle sound and absence of a reward sound, without any negative or “death-like” visual effects, to ensure suitability for pediatric participants.

Both environments were designed to be compatible with the OpenXR norm (Khronos Group, Beaverton, OR, USA). Users navigated and interacted with the virtual reality scene using controllers, which provided precise control over their movements and enhanced their sense of immersion [19].

### 2.3. Measurements

#### 2.3.1. Heart Rate Variability (HRV) to Assess the Autonomic Nervous System

Heart rate variability (HRV), generated from an electrocardiogram (ECG), can be used to evaluate the autonomic nervous system (ANS). HRV reflects the function of both the sympathetic and the parasympathetic systems [5].

The ECG signals were collected using Lead I, right arm or clavicle and left arm or clavicle, using the Dual Wireless ECG BioNomadix module pair (BIOPAC Systems, Inc., Goleta, CA, USA), consisting of a matched transmitter and receiver pair optimized for electrocardiogram data (https://www.biopac.com/product/bionomadix-rsp-with-ecg-amplifier (accessed on 1 March 2025)). The high signal-to-noise ratio and high time-based sampling resolution permit the pair to be used for exacting heart rate variability studies. Physiological signal data is transmitted at a rate of 2000 Hz, providing high-resolution waveforms at the receiver module’s output. These units interface with the MP160 (USA, Biopac Systems Inc., Goleta, CA, USA) data acquisition and analysis platform and AcqKnowledge 5.0 software (BIOPAC Systems Inc., Santa Barbara, CA, USA). AcqKnowledge software automatically scores the data and extracts the measurements of interest on a cycle-by-cycle basis. AcqKnowledge also includes a fully automated HRV analysis feature. The HRV parameters were analyzed offline via Kubios HRV software version 2 (https://www.kubios.com/ (accessed on 1 March 2025)) to measure the change in HRV. The time domain analysis of HRV included Mean RR, or the mean of RR intervals; the root mean square of successive differences between successive RR intervals (RMSSD), which reflects the beat-to-beat variance in HR and estimates the vagally mediated changes reflected in HRV; the standard deviation of NN intervals (SDNN); the proportion derived by dividing NN50 by the total number of NN intervals (pNN50); and the root mean square of successive differences of NN intervals (RMSSD), representing parasympathetic activity. The frequency domain analyses included the absolute and normalized powers’ values of low frequency (LF) and high frequency (HF), the LF/HF ratio, indicating sympathovagal balance, following the recommendations of the Task Force of the European Society of Cardiology and the North American Society of Pacing and Electrophysiology [20]. Psychological stressors are associated with decreased SDNN and an increase in the LF/HF ratio, which is characterized by decreased vagal tone and increased sympathetic activity [21].

#### 2.3.2. Psychological Measurements

To assess different dimensions of psychological stress and eating behavior, we employed a combination of trait-level and state-level questionnaires. Prior to the VR exposure, participants completed the Perceived Stress Scale (PSS), which captures chronic perceived stress over the previous month, and the Three-Factor Eating Questionnaire (TFEQ), which measures trait-like dimensions of eating behavior: cognitive restraint (CR), uncontrolled eating (UE), and emotional eating (EE). Following the VR stimulation, participants completed visual analog scales (VAS) designed to capture acute, momentary states of stress, anxiety, appetite, and craving. This approach enabled us to evaluate both stable behavioral tendencies and immediate subjective responses to the experimental paradigm.


*Stress and Anxiety Assessment*



*Pre-exposure*


Perceived Stress Scale (PSS). To assess baseline stress perception before VR, the Perceived Stress Scale (PSS) designed by Cohen [22,23] and adapted for children and adolescents was applied. The PSS consists of 10 items.


*Post-exposure*


Stress and Anxiety Assessment (VAS Method):

Immediately after the VR exposure, participants rated their perceived levels of stress and anxiety using two items: “How stressed are you feeling right now?” and “How anxious are you feeling right now?” Responses were recorded using visual analogue scales (VAS), ranging from 1 (“Not at all”) to 10 (“Extremely”), allowing for a subjective quantification of emotional response.


*Eating Behavior Questionnaires*



*Pre-exposure*


Three-Factor Eating Questionnaire for Children (TFEQ-R21C)

TFEQ measures the individual differences in eating patterns [24]. The TFEQ-R21 asks participants to respond to 21 questions on a four-point Likert scale for items 1–20 and on an eight-point numerical rating scale for item 21. Responses to each of the items are given a score between 1 and 4. Before calculating domain scores, items 1–16 were reverse-coded, and item 21 was recoded as follows: 1–2 scores as 1; 3–4 as 2; 5–6 as 3; 7–8 as 4. Domain scores were then calculated as a mean of all items within each domain; hence, domain scores also ranged from 1 to 4: cognitive restraint (CR)-factor, six items; uncontrolled eating (UE)-factor, nine items; and emotional eating (EE)-factor, six items, with higher scores being indicative of greater CR, UE, and EE [25].


*Post-exposure*


Appetite and Craving Ratings (VAS Method)

Immediately after the VR exposure, participants assessed their post-exposure appetite using two items: “How strong is your desire to eat right now?” and “How strong is your desire to eat sweets right now?” These responses were recorded using visual analogue scales (VAS), ranging from 1 (“Not at all”) to 10 (“Extremely”), describing the subjective feelings of hunger and cravings for sweets [26].

#### 2.3.3. Saliva Collection and Measurement of Stress Markers

Saliva samples were collected at three time points, 10 min before the VR exposure (T − 10 min), immediately after the exposure (T + 10 min), and 35 min after the onset of the exposure (T + 35 min), corresponding to the expected peak in salivary biomarker response. The VR exposure began at T0 min. Saliva collection was performed using Salivette^®^ devices (Sarstedt, Nümbrecht, Germany). A neutral cotton-based swab was placed into the mouth for about 2 min, allowing it to absorb the saliva. The saturated swab was introduced to the suspended insert, and the tube was closed firmly. The Salivette^®^ tubes were kept at −80 °C for storage until analysis could be performed. On the day of the assay, Salivette^®^ samples can be thawed and spun at low speed to separate the saliva sample from the gauze.

Salivary cortisol. The salivary cortisol was measured using a commercial direct immunoenzymatic method for salivary cortisol, KAPDB290 Cortisol Saliva ELISA (Diametra, Spello, Italy), with 0.033 ng/mL sensitivity and an analytical range of 0.5–50 [18].

Salivary alpha-amylase. The salivary cortisol was measured using a commercial direct immunoenzymatic method for salivary alpha-amylase, E-EL-H0320 Human AMY1(Amylase Alpha 1, Salivary) (Elabscience, Houston, TX, USA), with 0.94 ng/mL sensitivity and an analytical range of 1.56–100 [18].

#### 2.3.4. Anthropometric Measurements

Anthropometric measurements (weight, waist, and abdominal circumference) were obtained from the patient’s file. Actual BMI (kg/m^2^) was calculated by dividing weight (kg) by height squared (m^2^). BMI was classified based on World Health Organization (WHO) guidelines [15].

### 2.4. Virtual Reality (VR) Game-Based Experimental Model

The VR game exposure was selected as a low-intensity, emotionally neutral task to mildly activate the autonomic nervous system. This was intentional for ethical considerations in pediatric participants. Unlike traditional psychosocial stressors, this protocol was not designed to provoke strong emotional or evaluative stress, but rather to serve as a controlled, immersive stimulus for evaluating physiological reactivity in children and adolescents with obesity.

The participants were recruited during day hospitalization visits at Center for Diagnosis, Counseling, and Monitoring of Obese Children at the Emergency Clinical Hospital for Children Sfanta Maria Iaşi. All experimental sessions were conducted between 11:00 and 14:00, a period selected to minimize the influence of early morning cortisol peaks while ensuring participant availability during school hours. To reduce variability in appetite and metabolic parameters, a standardized 2 h fasting period was required prior to testing. The children were tested individually for a total duration of 55 min. The children’s parents or tutors approved and signed an online informed consent document.

The investigation was conducted by the same examiner (C.A.O.). Before the experiment, the principal investigator provided an overview of the study, and all the participants were given informed consent to be signed online. The experiment had the following aims: 1. Exposure to virtual environments. 2. Collecting saliva and analyzing the salivary stress markers. 3. Measuring with the help of wireless transmitters fixed on the skin of the electrocardiogram signals during the virtual environment exposure. 3. Completing questionnaires regarding eating behavior and stress perception before and after virtual environment exposure.

#### 2.4.1. Pre-Exposure Stage (T − 20 min to T0 min)

Participants were asked to fast for 2 h before arrival. Upon arrival, they provided informed consent and completed the Perceived Stress Scale (PSS) and Three-Factor Eating Questionnaire for Children (TFEQ-R21C). At T − 10 min, the first saliva sample was collected (baseline).

Subsequently, ECG sensors were attached (BIOPAC system), followed by a 5 min calibration and a 5 min HRV baseline recording (T − 5 min to T0 min). Participants were then fitted with Oculus VR headsets and provided with standardized instructions.

#### 2.4.2. VR Exposure Stage (T0 min to T + 10 min)

This stage consisted of two 5 min sequential VR conditions:

*Relaxation Phase (T0 min to T + 5 min):* Participants were immersed in a calming virtual forest environment (Forest app) to induce a parasympathetic response. ECG was recorded during this period, but no saliva sample was collected immediately afterward.

*Stimulation Phase (T + 5 min to T + 10 min):* Participants played an interactive VR game designed to stimulate cognitive and emotional engagement. ECG continued during this phase. At T + 10 min, immediately after VR exposure, the second saliva sample was collected.

#### 2.4.3. Post-Exposure Stage (T + 10 min to T + 35 min)

After VR exposure, participants completed the post-exposure PSS and VAS scales for appetite, sweet craving, stress, and anxiety. To capture the delayed endocrine response, the third saliva sample was collected at T + 35 min, corresponding to the expected peak in salivary cortisol and alpha-amylase. No ECG recordings were performed during this post-exposure phase.

### 2.5. Statistical Analysis

Data were analyzed using the Statistical Package for the Social Sciences (SPSS), version 22 (IBM Corp., Armonk, NY, USA). This study was designed as an exploratory pilot investigation. The final sample included 30 participants, which is within the range recommended for pilot studies and consistent with prior exploratory research involving heart rate variability and salivary biomarkers. The distribution of continuous variables was assessed using the Shapiro–Wilk test, which is more appropriate for small sample sizes (n < 50).

Due to the small sample size and non-normal distribution of most variables, continuous data are reported as median and interquartile range (IQR: Q1–Q3). Categorical variables are presented as frequencies and percentages. To compare pre- and post-exposure measurements of heart rate variability (HRV) parameters and salivary stress biomarkers, the Wilcoxon signed-rank test was used, based on the non-parametric nature of the data. Exploratory factor analysis (EFA) was conducted using principal component extraction (PCA) with oblimin (oblique) rotation, as our aim was to reduce data dimensionality and identify potentially correlated latent components underlying autonomic and endocrine reactivity (%Δ physiological variables). Sampling adequacy was assessed using the Kaiser–Meyer–Olkin (KMO) test, which yielded a value of 0.671, and Bartlett’s test of sphericity, which was significant (χ^2^ = 61.9, df = 15, *p* < 0.001), confirmed the suitability of the data for factor analysis. Based on eigenvalues >1 and the scree plot, three factors were retained. Regression-based factor scores (Regr scores) were then computed in SPSS for each participant to quantify individual expression of the identified factors, which were used in subsequent correlational analyses. These factor scores were subsequently used in bivariate Spearman correlation analyses to assess associations with the Three-Factor Eating Questionnaire (TFEQ-R21C) subscale scores—cognitive restraint (CR), uncontrolled eating (UE), and emotional eating (EE)—as well as with subjective stress and appetite ratings following VR exposure.

## 3. Results

### 3.1. Participant Characteristics

Table 1 summarizes the demographic and anthropometric characteristics of the participants.

### 3.2. Autonomic Nervous System Responses to VR Game-Based Experimental Model

Table 2 presents the changes in autonomic nervous system activity, as reflected by heart rate variability (HRV) parameters, in children with obesity following exposure to virtual reality (VR) environments. Although mean RR interval and heart rate (HR) are mathematically inverse, RR was selected as the primary variable to reflect cardiac chronotropic activity and vagal modulation. RR intervals provide more precise beat-to-beat information and are commonly used in HRV research to assess autonomic responses. A Wilcoxon signed-rank test comparing responses to two VR conditions—(1) a calming forest environment (relaxation) and (2) an interactive game application (stimulation)—revealed statistically significant changes. Specifically, exposure to the VR game app resulted in a significant decrease in both the mean RR interval and the root mean square of successive differences (RMSSD), along with a significant increase in mean heart rate (HR) and the low-frequency to high-frequency (LF/HF) ratio, relative to the forest environment (Figure 3). These findings suggest a shift toward increased sympathetic activity during the stimulating VR condition. No significant differences in HRV responses were observed between sexes or across age groups, as assessed using the Mann–Whitney U test.

### 3.3. The Salivary Stress Markers’ Responses to VR Game-Based Experimental Model

To capture the endocrine stress response more accurately, we focused on the percent change (%Δ) from post-relaxation (T + 10 min) to peak response (T + 35 min), as this interval reflects the expected latency of HPA and SAM axis responses to acute stimulation. This approach reduces inter-individual baseline variability and has been used in similar psychoneuroendocrinological studies [28].

Table 2 presents the changes in salivary stress marker levels at two post-exposure time points, T + 10 min (immediately following the VR game exposure) and T + 35 min (25 min after the VR game, corresponding to the expected peak in salivary biomarker response). A Wilcoxon signed-rank test comparing the two time points revealed a statistically significant increase in salivary alpha-amylase. No significant change was observed in salivary cortisol levels between T + 10 and T + 35 min (Figure 3).

In the Figure 4, the boxplots illustrate the median and 95% confidence intervals (CIs) of the percentage change (%Δ) in heart rate variability (HRV) parameters and salivary stress markers (alpha-amylase and cortisol) following VR game-based exposure. These plots represent the central tendency and variability of autonomic and neuroendocrine responses, capturing individual differences in physiological reactivity among children and adolescents with obesity.

### 3.4. Subjective Stress and Eating Responses to VR Game-Based Experimental Model

#### 3.4.1. Perceived Stress Response

*Perceived Stress Scale (PSS)*. Perceived stress was assessed using the Perceived Stress Scale (PSS), and baseline scores prior to VR exposure are presented in Table 2. Elevated stress levels, defined as scores above the 75th percentile, were observed in eight participants (26.6%)

*Stress and Anxiety Assessment (VAS Method).* Participants rated their current feelings of stress and anxiety using two visual analogue scale (VAS) questions ranging from 1 (“Not at all”) to 10 (“Extremely”). Due to the non-normal distribution of VAS scores, medians and interquartile ranges (IQRs) are reported (see Table 2).VAS scores for stress and anxiety ratings showed a left-skewed distribution, with a subset of participants reporting high stress ratings (VAS Stress > 70, n = 1, 3.3%) and high anxiety ratings (VAS Anxiety > 70, n = 2, 6.7%).

#### 3.4.2. Eating Behavior Response

*Appetite and Craving Ratings (VAS Method).* Table 2 presents the median and interquartile range (IQR: Q1–Q3) for responses to appetite-related VAS items administered after the VR game. Due to the non-normal distribution of VAS scores, medians and interquartile ranges (IQRs) are reported (see Table 2). VAS scores for appetite and craving ratings showed a left-skewed distribution, with a subset of participants reporting high hunger ratings (VAS Appetite > 70, n = 7, 23.3%) and high craving ratings (VAS Craving > 70, n = 2, 6.7%).

### 3.5. Physiological Stress Reactivity

Normoreactive vs. Hyporeactive Patterns

Based on predefined thresholds (≥10% for HRV, ≥20% for salivary markers) [29,30], the majority of children showed expected autonomic stress responses, while endocrine responses were less consistent (Table 3).

This suggests a predominantly sympathetic autonomic response (HRV-based), while HPA axis responses (cortisol) were blunted or minimal in most participants.

### 3.6. Factor Analysis and Associations with Eating Patterns

To identify latent patterns in physiological stress responses following VR game exposure, an exploratory factor analysis was conducted without salivary cortisol, due to the observed blunted response, using principal component extraction with oblimin rotation on the percentage change (%Δ) of salivary alpha-amylase and HRV parameters (Table 4, Figure 5). The analysis yielded two interpretable components, cumulatively accounting for 69.1% (51.3%, 17.8%, respectively).

These two components represent distinct autonomic response patterns: parasympathetic modulation (Factor 1) and sympathetic activation (Factor 2).

*Factor 1—Parasympathetic Reactivity*: Characterized by strong positive loadings for RMSSD and SDNN reflecting classic time-domain parasympathetic HRV markers. Moderate negative loading on LF/HF suggests lower sympathetic dominance, associated with stronger parasympathetic modulation. This factor represents a shift toward parasympathetic dominance and higher HRV. It reflects autonomic flexibility and adaptive cardiac vagal engagement [5,31,32].

*Factor 2—Sympathetic Activation*: Driven by increases in salivary alpha-amylase (sAA), a salivary marker of sympathetic-adrenal activation, and increases in heart rate (HR), both indicators of stress-induced sympathetic dominance. Moderate negative loading on RR suggests shortened inter-beat intervals, consistent with increased heart rate. This factor represents a classic sympathetic stress reactivity pattern [5,31,32].

### 3.7. Spearman Correlations Between Physiological Response Factor Scores and Eating Pattern Measures

These factors were used in further statistical models (e.g., correlation with stress, appetite, or eating behavior) to assess how autonomic regulation patterns relate to psychobehavioral outcomes in children with obesity.

Table 5 presents the Spearman rank correlation analysis between the regression-based factor scores (derived from exploratory factor analysis) and self-reported stress and eating behavior measures.

The parasympathetic reactivity score (Factor 1) showed a significant negative correlation with PSS (pre-exposure) (r = −0.365, *p* = 0.05) and VAS anxiety (r = −0.430, *p* = 0.02), and a significant positive correlation with uncontrolled eating (UE) (r = 0.510, *p* = 0.004). These findings suggest that vagal reactivity was linked to lower reported anxiety and a tendency toward dysregulated eating. The sympathetic activation score (Factor 2) showed a positive correlation with VAS anxiety (r = 0.470, *p* = 0.02) and cognitive restraint (CR) (r = 0.482, *p* = 0.02), indicating that greater sympathetic arousal was associated with higher subjective stress and a stronger tendency to consciously control eating (CR).

Other correlations (cross-correlations): Pre-exposure PSS was positively correlated with Central Obesity Index (r = 0.518, *p* < 0.01). BMI was positively correlated with emotional eating (EE) (r = 0.566, *p* < 0.01), VAS sweet craving (r = 0.350, *p* = 0.05), VAS stress (r = 0.466, *p* < 0.001), and VAS anxiety (r = 0.458, *p* = 0.01). Uncontrolled eating (UE) was positively correlated with emotional eating (EE) (r = 0.449, *p* = 0.02) and VAS appetite (r = 0.655, *p* < 0.001) and negatively correlated with cognitive restraint (CR) (r = 0.383, *p* = 0.04).

## 4. Discussions

The prevalence of overweight and obesity has increased worldwide. There is a relationship between maladaptive eating patterns (e.g., binge eating, emotional eating, unhealthy dietary restraint) and emotional functioning among patients with obesity and non-clinical patients with excessive body weight. Emotional dysregulation is defined as difficulty in receiving, processing, and displaying emotions and a lack of adaptive coping with emotions and stress [33].

To investigate the relationship between emotional dysregulation and eating behavior in children and adolescents with obesity, experimental stress paradigms can be employed to reveal potential dysregulations in autonomic and neuroendocrine responses. While the Trier Social Stress Test (TSST) is the gold standard for eliciting cognitive, emotional, and behavioral stress, alternative models, such as immersive VR-based game exposure, have also been successfully used in pediatric populations to simulate ecologically valid stress conditions.

Our study investigated the cardiac autonomic function in children and adolescents with obesity by analyzing their physiological responses to a virtual reality (VR) social-emotional stressor. The objective of the study was to identify patterns of autonomic and neuroendocrine reactivity using exploratory factor analysis, and to explore associations between these physiological patterns and perceived stress, anxiety, and eating behavior in children and adolescents with obesity.

The use of both trait-level (PSS, TFEQ) and state-level (VAS) measures allowed us to differentiate between stable psychological tendencies and acute subjective responses following VR game exposure. This separation is especially important in pediatric populations, where contextual sensitivity and trait–state interactions influence emotional regulation and eating behavior. This differentiation is supported in the literature. Prior researches emphasize the importance of using real-time or momentary assessment tools, such as VAS, to evaluate state responses, especially in studies on emotional regulation and eating behavior [34,35,36].

### 4.1. VR Game-Based Stimulation [35]

In our study, we used VR game exposure to elicit physiological reactivity, consistent with emotion- and social-stressor models. Previous studies showed that virtual exposure to certain games can induce stress by activating different parts of the brain, which play significant roles in emotions and cognition [13].

The experimental design included two sequential phases, an initial stress-reduction phase, followed by a psychological stress-induction phase. Participants first engaged in a 5 min immersion in a relaxing virtual environment using the Forest application, followed by a 5 min cognitively stimulating VR game intended to induce acute psychological stress. Stress responses were assessed using both subjective ratings and physiological indicators. Autonomic nervous system (ANS) activity was evaluated using heart rate variability (HRV), while salivary biomarkers, cortisol (HPA axis), and alpha-amylase (sympathetic nervous system), were used to capture neuroendocrine responses [37,38].

It is important to note that our VR game-based stimulation was intentionally designed as a mild, non-threatening paradigm to elicit physiological reactivity without invoking the strong psychosocial stress components typically found in protocols like the Trier Social Stress Test (TSST).

Several studies have demonstrated that the VR-TSST effectively elicits subjective stress and autonomic responses comparable to those observed in the traditional TSST [39,40,41,42,43]. However, VR-TSST protocols tend to elicit blunted HPA-axis activation, as evidenced by attenuated salivary cortisol responses in comparison to the traditional TSST [41,42,44]. These findings suggest that while immersive VR scenarios succeed in triggering real-time stress appraisal and sympathetic nervous system reactivity, they may lack the full social-evaluative threat required to robustly stimulate cortisol secretion, possibly due to reduced perceived presence or emotional salience in virtual settings.

A recent meta-analysis comparing traditional and VR-TSST protocols concluded that while subjective and cardiovascular stress responses were largely equivalent, cortisol reactivity was consistently lower in VR-based settings, emphasizing the nuanced dissociation between autonomic and neuroendocrine stress pathways [40]. Repeated exposure to both real and VR-TSSTs further supports this dissociation, showing faster habituation in cortisol responses in virtual conditions [42]. The refined version of VR-TSST, featuring improved immersion, interactive audiences, and realistic social dynamics, showed more robust endocrine responses [43].

### 4.2. Autonomic and Endocrine Responses to VR Game-Based Experimental Model

Variations in heart rate (HR) responses to a VR game reported across studies may be primarily influenced by the competitive intensity of the gaming context rather than the specific game title [16,17]. The researchers emphasized the role of perceived competitiveness in modulating cardiovascular responses (increased heart rate) when the participants were exposed to a competitive gaming environment. In terms of heart rate variability (HRV), reductions in RMSSD have been consistently observed during gameplay, reflecting decreased vagal modulation. While SDNN often remains unaffected, other parameters such as low- and high-frequency HRV components show meaningful changes. For example, Yeo et al. found a significant increase in low-frequency power, a concurrent decrease in high-frequency power, and an elevated LF/HF ratio during a session of League of Legends (LoL), indicating a shift toward sympathetic dominance and reduced parasympathetic activity [17,18].

In our study, 30 children and adolescents were exposed to a VR game-based stimulation while heart rate (HR), heart rate variability (HRV), and salivary cortisol and alpha-amylase (sAA) were also collected to assess autonomic and neuroendocrine activation. Participants completed validated self-report measures on perceived stress, anxiety, and eating behaviors, including cognitive restraint, uncontrolled eating, and emotional eating.

In this study, physiological reactivity to immersive VR game exposure was assessed using the percentage change (%Δ) from baseline. For HRV parameters, a response was considered “normoreactive” if it involved a decrease ≥10% in RR intervals, SDNN, RMSSD, or pNN50, and an increase ≥10% in LF/HF ratio. For salivary markers, a response was considered “normoreactive” if there was an increase ≥20% in salivary cortisol and/or alpha-amylase levels post-exposure [29,30]. Participants who did not meet these thresholds were categorized as “hyporeactive”. Based on these predefined physiological thresholds, 100% of participants demonstrated a normoreactive decrease in RR intervals (≥10%), while 70% showed a sympathetic shift, as reflected in LF/HF increases. For salivary biomarkers, 13% exhibited a ≥20% increase in cortisol, and 93% showed an increase in alpha-amylase, indicating a blunted cortisol response. This pattern suggests a robust autonomic response paired with a blunted hypothalamic–pituitary–adrenal (HPA) axis response, a response frequently reported in children with obesity. These findings support the validity of the VR game-based experimental model, as evidenced by significant reductions in HRV indices (RR, RMSSD, SDNN) and increases in LF/HF ratio. The limited cortisol reactivity may reflect both the attenuated HPA axis responsiveness characteristic of pediatric obesity [45,46] and the moderate psychological challenge of the VR stressor used in this study [42].

The blunted cortisol response observed in our study aligns with existing evidence indicating hypothalamic–pituitary–adrenal (HPA) axis dysregulation in individuals with obesity. Research has shown that obesity, particularly with central fat accumulation, is associated with attenuated cortisol responses to stress. For example, Therrien et al. [28] demonstrated that both obese and weight-reduced adults exhibited a significantly reduced cortisol awakening response (CAR) compared to lean individuals, with gender and visceral fat distribution being key moderating factors. Similarly, Hillman et al. [47] reported diminished HPA axis reactivity among adolescent females with obesity, suggesting that early-onset obesity may impair neuroendocrine stress responsiveness during critical developmental periods. These findings support the hypothesis that chronic metabolic and inflammatory alterations in obesity contribute to a downregulation of cortisol output, potentially as an adaptive or maladaptive response to repeated stressor exposure. Similarly, Herhaus et al. [48] found that individuals with obesity showed lower cortisol reactivity to psychosocial stress, which was associated with increased food intake, suggesting a dysregulated endocrine response that may facilitate non-homeostatic or emotionally driven eating behaviors. Supporting these findings in younger populations, our previous research [45] using a digital adaptation of the Trier Social Stress Test demonstrated attenuated cortisol responses in overweight adolescents, along with significant associations between cortisol reactivity and stress-related eating patterns.

In this part of the study, we have highlighted how VR game-based stimulation can elicit individual autonomic and endocrine responses in children and adolescents with obesity. Monitoring autonomic and endocrine responses under stress in youth with weight excess can reveal different emotional eating patterns, and these could reveal early indicators of metabolic or cardiovascular vulnerability.

There is no clear consensus regarding the presence or pattern of autonomic dysfunction in children and adolescents with obesity, as studies report varying findings, ranging from reductions to increases, or even no significant changes, in heart rate variability (HRV) indices [40,41,42,43]. However, a recent meta-analysis by Papadopoulos et al. [42], which included 12 studies of participants aged 5 to 18 years (mean age 10.5 years), provided evidence for the presence of autonomic imbalance in pediatric obesity. Specifically, obese children and adolescents exhibited significantly lower vagal activity, as indicated by reduced high-frequency (HF) power, RMSSD, and PNN50. Several studies have emphasized that the variability in autonomic findings may be partially explained by differences in the duration of obesity. According to Rabbia et al. [43], children with shorter durations of obesity typically exhibit reduced parasympathetic tone across time- and frequency-domain parameters, accompanied by increased sympathetic activity. On the other hand, those with longer durations of obesity tend to show persistent reductions in parasympathetic function, particularly in time-domain measures, along with increased LF/HF ratios, reflecting a shift toward sympathetic dominance.

Further, a brief stressor can reveal a subtle autonomic dysfunction. Studying the emotional eating behavior and cardiac autonomic modulation during a short-duration physical challenge in adolescents, González-Velázquez et al. [49] provide insight into stress reactivity and its links to eating behaviors in adolescents. The 2 min Sustained Weight Test (SWT) revealed a marked vagal decline, especially in adolescents with high emotional eating scores. SDNN and RMSSD decreased under stress in these individuals, indicating a vagal imbalance not visible at rest. Other studies have investigated autonomic responses to acute stress, reinforcing the importance of parasympathetic recovery. Herhaus et al. [48] investigated the autonomic response in individuals with obesity and with a healthy weight during a standardized psychosocial stress, the Trier Social Stress Test (TSST). The findings suggest that while obese and healthy-weight individuals may respond similarly to acute stress in terms of autonomic suppression, those with obesity exhibit reduced vagal rebound during recovery, demonstrating significantly lower RMSSD and marginally lower HF-HRV compared to healthy-weight controls. The authors suggested that this autonomic recovery pattern may reflect diminished parasympathetic flexibility, which is relevant for understanding obesity-related risks for cardiovascular and stress-related disorders.

These studies highlighted HRV parameters, such as RMSSD, as a sensitive indicator of autonomic function in youth. Obesity is associated with reduced parasympathetic activity both at rest and under stress, while physical or emotional stressors often reveal vagal withdrawal not apparent in baseline measures.

### 4.3. Correlations Between Physiological Response Factor Scores and Eating Pattern Measures

Our data revealed significant positive correlations between BMI and subjective perceptions of stress, anxiety, and appetite, as well as between pre-exposure PSS scores and the Central Obesity Index. These findings support the hypothesis that chronic perceived stress and heightened appetitive drive are closely linked to excess weight and central adiposity in children with obesity. This pattern reinforces emotion-driven models of eating behavior, in which stress contributes to dysregulated eating and reduced dietary control [46,50].

Three abnormal eating patterns were assessed in the present research: emotional, uncontrolled, and restrictive. Each of these eating styles contains a pathological aspect: (1) unhealthy restriction of food intake (restrictive eating), (2) loss of control over the rate and amount of food consumed (uncontrolled eating), and (3) too frequent eating under emotions and stress (emotional eating) [33].

Our results indicated that immersive virtual reality (VR) game exposure may represent a valid experimental model to examine the link between stress reactivity and eating behavior in children and adolescents with obesity. In our study, a brief stressor revealed that children with obesity could have individual stress responses that are associated with dysfunctional eating behavior. Through exploratory factor analysis, we identified two distinct patterns of physiological response.

The first pattern, parasympathetic reactivity factor, was characterized by increased RMSSD and SDNN, and a decreased LF/HF ratio, indicating enhanced vagal activity and greater autonomic flexibility. This pattern was associated with lower perceived stress and anxiety, but paradoxically also with higher uncontrolled eating. The parasympathetic reactivity factor appears to capture adaptive vagal engagement, where higher parasympathetic modulation is associated with lower subjective emotional stress and anxiety. This pattern reflects a resilient autonomic profile, children with greater vagal modulation experience less subjective stress and anxiety. This aligns with Thayer and Lane’s neurovisceral integration model [32] and polyvagal theory [51], which describes vagal tone as a marker of adaptive emotional regulation [52]. According to the polyvagal theory [51], higher vagal tone not only supports social engagement and emotion regulation but may also amplify awareness of internal states. In individuals with low cognitive restraint or impaired executive functioning (common in pediatric obesity), this heightened interoceptive sensitivity may contribute to disinhibited or compulsive eating in response to stress or food-related cues. Some studies have associated reduced parasympathetic activity in adolescents with difficulty with behavioral control, potentially contributing to overeating tendencies [43,47]. In our VR game paradigm, the parasympathetic reactivity pattern was positively associated with uncontrolled eating, suggesting a paradox: although children with stronger parasympathetic reactivity appear more emotionally resilient, they may also be more sensitive to internal cues (e.g., hunger, cravings), especially in the absence of adequate cognitive control. This interpretation is supported by findings from behavioral medicine, which suggest that high vagal tone can enhance interoceptive awareness, potentially facilitating cue-driven eating in individuals with poor cognitive regulation [46,51].

The second pattern, sympathetic activation, was characterized by increased heart rate (HR) and salivary alpha-amylase, as well as decreased respiratory rate (RR) intervals, indicating sympathetic arousal. It correlated with greater cognitive restraint. The sympathetic activation factor may indicate that children who exhibit stronger cardiac autonomic reactivity are more aware of stress and may engage in conscious restraint to avoid overeating (compensatory control). This aligns with laboratory studies of stress-induced eating behavior, which reported that high dietary restraint individuals have greater salivary cortisol levels than low restraint individuals [53].

The studies aimed at investigating the relationship between restrained eating behavior, resistance to distractor interference when exposed to food cues, and cardiac autonomic regulation confirmed that restrained eating was associated with sympathetic dominance that may reflect self-regulatory deficiencies of restrained eaters [54]. Wu et al. investigated the relationship between high-frequency heart rate variability (HF HRV) and food craving in healthy adolescents. Results showed that lower HF HRV was significantly associated with higher food craving, particularly in the lack of control over eating subscale [55]. Previous research indicated that stress-induced sympathetic activation can be accompanied by maladaptive eating behaviors, including restrictive or disinhibited eating, depending on cognitive and emotional regulation capacity. Some children may exhibit higher CR in response to a perceived threat, as a top-down effort to maintain dietary control despite internal arousal (compensation via food intake restraint) [56,57]. Restrained eaters are often unsuccessful in maintaining their cognitive control and tend to have a higher BMI and to experience weight cycling. One explanation for this association could be that restrained eaters are only temporarily able to maintain their cognitive restriction over their eating behavior and demonstrate disinhibited food intake in a variety of experimental situations involving food exposure. Moreover, studies investigating the relationship between restrained eating, attention to food cues, and autonomic regulation found that restrained eaters show greater sympathetic dominance, which may reflect dysregulation in self-regulatory systems. For example, Meule et al. demonstrated that restrained eating is associated with faster reactions to high-calorie food cues and altered cardiac autonomic balance, suggesting impaired top-down inhibitory control in response to tempting stimuli [54]. Similarly, Wu et al. found that lower high-frequency HRV (a marker of reduced parasympathetic tone) was significantly associated with higher food cravings in adolescents, particularly in those with difficulties in controlling food intake—further supporting the link between autonomic dysregulation and eating vulnerability [55]. From a neurobiological perspective, stress-induced sympathetic activation may promote both restrictive and disinhibited eating patterns, depending on the individual’s cognitive control and emotional regulation capacity. Some children may respond to internal arousal and perceived threat by increasing cognitive restraint, using top-down control strategies to suppress food intake [56,57]. However, restrained eating is often ineffective in the long term; such individuals are prone to weight cycling and increased BMI, as cognitive control tends to break down in high-stress or food-rich environments, leading to episodes of disinhibited eating.

Notably, no significant associations were observed between either physiological stress reactivity factor (parasympathetic reactivity or sympathetic activation) or emotional eating (EE) scores. This finding suggests that, in this sample of children and adolescents with obesity, EE may be less influenced by acute autonomic stress responses and more shaped by other psychological or contextual factors, such as chronic emotional states, coping style, or environmental cues [50,58].

These findings suggest two distinct stress-eating phenotypes in children with obesity: (1) emotionally buffered but behaviorally reactive (vagal-driven) and (2) emotionally aroused but cognitively compensatory (sympathetic-driven). The presence of parasympathetic-linked eating vulnerability and sympathetic-linked restraint may represent distinct regulatory profiles in pediatric obesity, with implications for tailored intervention strategies.

### 4.4. Limitations

This study should be interpreted as a pilot investigation, given the relatively small sample size in relation to the complexity of the physiological and behavioral data collected. Although the multimodal design allowed for detailed characterization of autonomic and endocrine reactivity, as well as subjective and behavioral outcomes, the sample size limits the statistical power and generalizability of the findings. The use of exploratory factor analysis and correlation testing was intended to identify preliminary patterns of physiological stress reactivity and their associations with eating behavior traits, rather than to confirm specific hypotheses. Additionally, the absence of a control group and the cross-sectional design preclude causal inference.

Furthermore, while we employed both trait-level (e.g., PSS and TFEQ) and state-level (e.g., post-exposure VAS ratings) psychological measures, the absence of validated, pre–post state-specific instruments for all subjective variables (such as appetite and stress) limits our ability to comprehensively evaluate short-term emotional reactivity. Although VAS tools are commonly used to capture acute responses in pediatric settings, future studies should consider including standardized, validated state measures to better differentiate transient versus enduring psychological patterns.

Moreover, several potential confounding factors that may influence stress reactivity and eating behavior were not adequately controlled. These include participants’ sleep quality, habitual physical activity, and the broader psychosocial and dietary environment within the family context. Such variables could impact both autonomic and HPA axis functioning, as well as eating behaviors, and their omission represents a limitation to the internal validity of the study. Future research should aim to address these limitations by recruiting larger samples and incorporating control groups. In future studies, we recommend exploring age-by-gender interactions in stress reactivity and eating behavior, as developmental and hormonal factors may modulate these responses in important ways.

## 5. Conclusions

In conclusion, our findings highlight the heterogeneity of physiological regulation in children with obesity. While parasympathetic reactivity may reflect both adaptive (reduced anxiety) and maladaptive (increased UE) processes, sympathetic activation may support compensatory restraint in the face of heightened emotional arousal. Our results extend the application of biopsychophysiological models of eating behavior into immersive stress-induction contexts, such as VR-based paradigms, and suggest that these distinct autonomic profiles may require differential intervention strategies in pediatric obesity management.

Given the exploratory nature and modest sample size of this study, the findings should be viewed as preliminary evidence that requires validation in larger, more representative studies.

## Figures and Tables

**Figure 1 nutrients-17-02492-f001:**
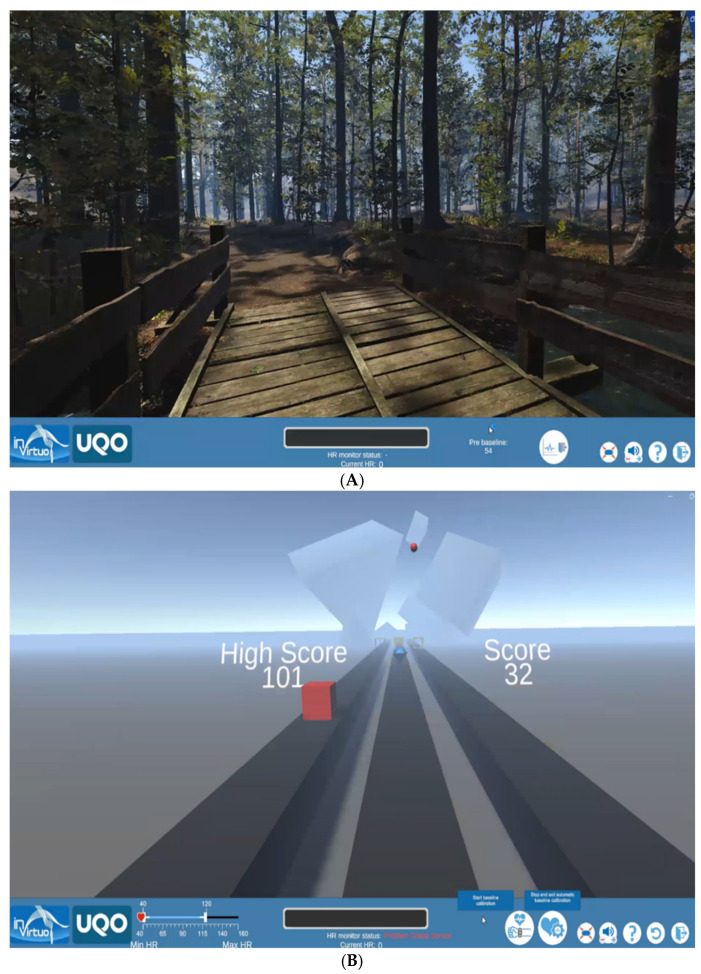
Screenshot from the VR applications. (**A**) Seated participant explores the VR Forest application. (**B**) Seated participant plays a virtual game.

**Figure 2 nutrients-17-02492-f002:**
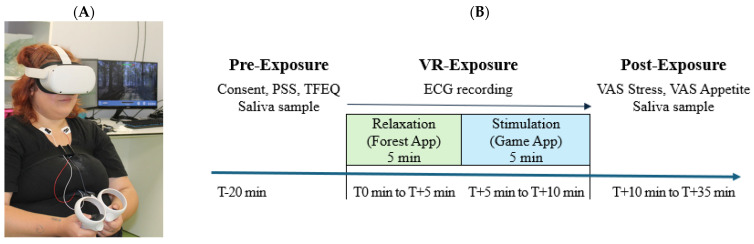
Study protocol. (**A**). Adolescent participant in the study. (**B**) Schematic summary of the procedure.

**Figure 3 nutrients-17-02492-f003:**
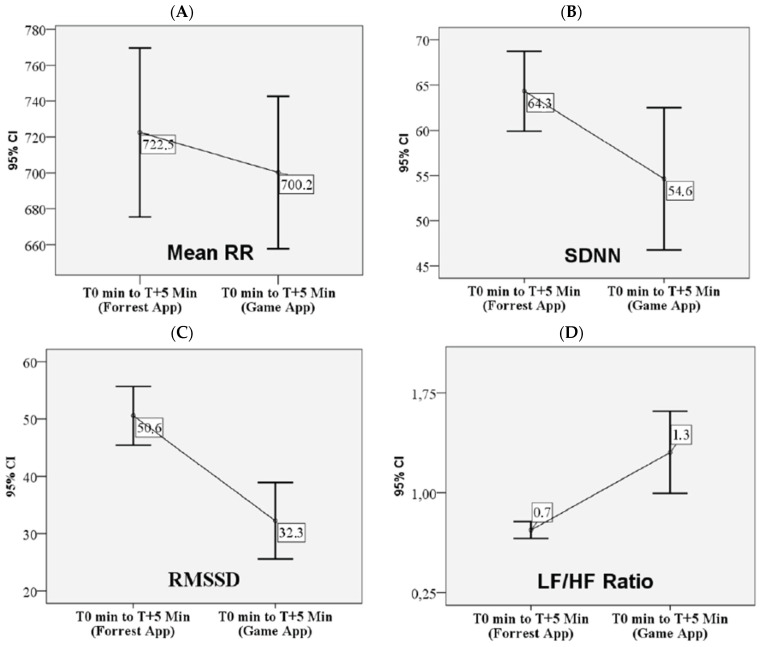
Effects of VR game exposure on heart rate variability (HRV): Mean RR (**A**), SDNN (**B**), RMSSD (**C**), and LF/HF Ratio (**D**), and salivary markers of stress cortisol (**E**) and alpha-amylase (**F**).

**Figure 4 nutrients-17-02492-f004:**
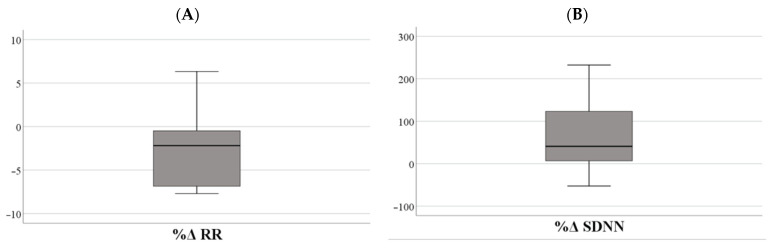
The boxplots illustrate the median and 95% confidence intervals (CIs) of the percentage change (%Δ) in heart rate variability (HRV) parameters and salivary stress markers following VR game-based exposure: %Δ RR (**A**), %Δ SDNN (**B**), %Δ RMSSD (**C**), %Δ LF/HF Ratio (**D**), and salivary markers of stress cortisol, %Δ sCortisol (**E**) and alpha-amylase, %Δ sAA (**F**).

**Figure 5 nutrients-17-02492-f005:**
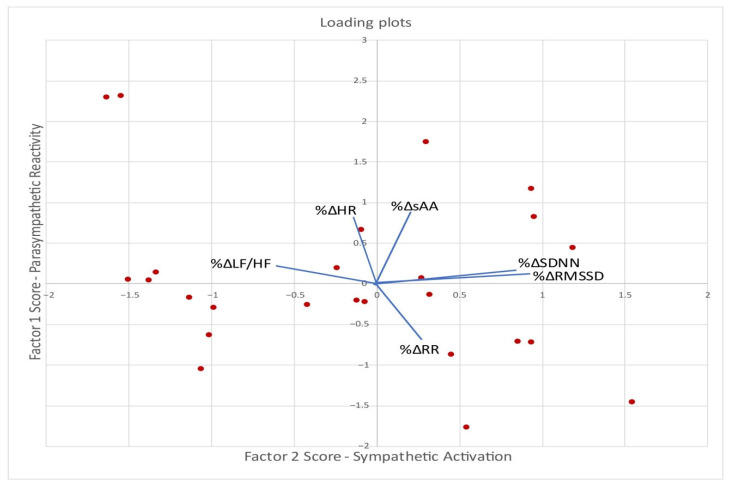
Principal component analysis biplot illustrating the effect of autonomic and neuroendocrine parameters in children and adolescents with obesity. The patients are represented as red dots.

**Table 1 nutrients-17-02492-t001:** Pre-exposure characteristics of the studied participants.

Characteristics	Participants (N = 30)
Age	
Age (years)	10 (9–14)
8–11 years (%)	56.7 (17)
12–17 years (%)	43.3 (13)
Anthropometric parameters	
Body Mass Index (BMI) (kg/m^2^)	27.6 (26.1–32.9)
Central Obesity Index	0.62 (0.58–0.65)
Severe obesity ^1^ (>3 SD) (%, N)	60 (18)

^1^ According to age-specific BMI cutoff points published by the WHO [27]. Due to the small sample size and the non-normal distribution of most variables, descriptive statistics are reported as median and interquartile range (IQR: Q1–Q3) for all continuous variables. Categorical variables are presented as frequencies and percentages.

**Table 2 nutrients-17-02492-t002:** Descriptive statistics of HRV measures, salivary stress markers, perceived stress, and subjective eating behavior as a response to the VR-exposure experiment.

Parameters	VR Game-Based Experimental Model		
Heart rate variability	T0 min to T + 5 min Relaxation Forest App	T + 5 min to T + 10 min Stimulation Game App	*p*-value
	ECG real-time monitoring		
Mean heart rate (min^−1^)	89 (80.5–92.3)	90 (85.3–97.3)	0.002
Mean RR (ms)	679 (632–814.3)	673 (616–785)	0.003
SDNN (ms)	65 (55.8–69)	60 (39.8–69.5)	0.026
RMSSD (ms)	47.5 (41–61)	25 (18.5–47.5)	0.000
PNN50 (%)	22 (14–31)	19 (14–28)	0.213
LF/HF ratio	0.8 (0.6–0.9)	1.3 (0.7–1.5)	0.001
Salivary stress markers	T + 10 min (Post VR)	T + 35 min (Peak-time)	*p*-value
	Saliva collection	Saliva collection	
Salivary cortisol (ng/mL)	22.4 (16.1–33.5)	20.6 (17.8–24.8)	0.6
Salivary alpha-amylase (ng/mL)	2.9 (2.2–6.1)	5.6 (3.8–6.8)	0.001
	Pre VR-exposure	Post VR-exposure	
Perceived stress	Questionnaires		
Perceived Stress Scale (PSS)	17.5 (13–23)		
Stress and Anxiety Assessment (VAS Method)			
How stressed are you feeling right now?	-	1 (1–2.3)	
How anxious are you feeling right now?	-	1 (1–4)	
Eating behavior parameters ^1^			
CR-factor	13 (11.8–18.2)	-	
UE-factor	21.5 (16–28.3)	-	
EE-factor	8 (6–10.2)	-	
Appetite and Craving Ratings (VAS Method)			
How strong is your desire to eat right now?	-	4.5 (2–6.3)	
How strong is your desire to eat sweets right now?	-	1 (1–3)	

^1^ Measured with the Three Factor Eating Questionnaire. Due to the small sample size and the non-normal distribution of most variables, descriptive statistics are reported as median and interquartile range (IQR: Q1–Q3) for all continuous variables. The Wilcoxon signed-rank test was used to compare two related samples. A *p*-value < 0.05 was considered statistically significant.

**Table 3 nutrients-17-02492-t003:** Autonomic and neuroendocrine reactivity to the VR game-based experimental model.

HRV Parameter/Salivary Marker/	Normoreactive (N, %)	Hyporeactive/No Response (N, %)
↓ RR ≥ 10%	30 (100%)	0 (0%)
↓ RMSSD ≥ 10%	26 (86.7%)	4 (13.3%)
↑ LF/HF ≥ 10%	21 (70%)	9 (30%)
↑ sCortisol ≥ 20%	4 (13.3%)	26 (86.7%)
↑ sAA ≥ 20%	28 (93.3%)	2 (6.7%)

↑ for increase, ↓ for decrease.

**Table 4 nutrients-17-02492-t004:** The pattern matrix obtained from exploratory factor analysis (EFA) was conducted using principal component extraction with oblimin rotations.

Variable	Component	
	1	2
%ΔRMSSD	+0.931	
%ΔSDNN	+0.869	
%ΔLF/HF	−0.583	
%ΔsAA		+0.839
%ΔHR		+0.800
%ΔRR		−0.645

Extraction Method: principal component analysis. Rotation method: oblimin with Kaiser normalization.

**Table 5 nutrients-17-02492-t005:** Spearman correlations between physiological response factor scores and eating pattern measures.

Variable	Parasympathetic Reactivity	Sympathetic Activation
	(Factor 1 *)	(Factor 2 *)
PSS (pre-exposure)	−0.365 (*p* = 0.05)	−0.135 (*p* = 0.48)
VAS Stress	−0.194 (*p* = 0.30)	0.322 (*p* = 0.08)
VAS Anxiety	−0.430 (*p* = 0.02)	0.470 (*p* < 0.01)
VAS Appetite	0.249 (*p* = 0.19)	−0.184 (*p* = 0.33)
VAS Sweet Craving	0.289 (*p* = 0.12)	−0.172 (*p* = 0.36)
Cognitive Restraint (CR)	−0.174 (*p* = 0.37)	0.422 (*p* = 0.02)
Uncontrolled Eating (UE)	0.553 (*p* < 0.01)	−0.225 (*p* = 0.23)
Emotional Eating (EE)	0.082 (*p* = 0.67)	0.185 (*p* = 0.33)

* Factor scores are regression-based scores extracted from exploratory factor analysis.

## Data Availability

The data presented in this study are available on request from the corresponding author. The data presented in this study are not publicly available due to privacy and ethical restrictions related to minors’ data.

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
