# Peer review of "Autonomic and Neuroendocrine Reactivity to VR Game Exposure in Children and Adolescents with Obesity: A Factor Analytic Approach to Physiological Reactivity and Eating Behavior"

_nutrients, 2025, doi:10.3390/nu17152492_

Round 1
Reviewer 1 Report
Comments and Suggestions for Authors
The submitted manuscript presents very interesting observations concerning the impact of stressful situation evoked by immersive virtual reality on the autonomic function and selected endocrine parameters in the group of children and adolescents with obesity. The design of the study is appropriate and the results are properly analyzed. The discussion is appropriate and presents the obtained data in the view o available literature. The results suggest that measurement of heart rate variability can be a marker of emotional resilience. Furthermore, immersive virtual reality stress exposure may become a valuable experimental model in assessment of stress reactivity among children with obesity.
Author Response
We are very grateful for the reviewer's positive assessment of all sections of our manuscript
Reviewer 2 Report
Comments and Suggestions for Authors
The paper “Autonomic and Neuroendocrine Patterns of VR-Induced Stress in Pediatric Obesity: A Factor Analytic Approach to Physiological Reactivity and Eating Behavior” explores the autonomic and neuroendocrine response patterns of obese children and adolescents and their association with eating behavior through a virtual reality (VR) -induced stress model. The research design is innovative and provides a valuable reference for understanding the relationship between childhood obesity and stress response as well as eating behavior. However, there are still some areas in the research that need improvement.
Comments:
Q1. The sample size of this project is relatively small, which may limit the statistical power and extrapolation of the results. It is suggested to supplement the explanation of the basis for estimating the sample size and clearly point out the possible limitations brought by a small sample in the discussion.
Q2. Although VR games have been mentioned as capable of inducing social emotional stress, there has been no systematic comparison of the effects of this model with those of traditional stress paradigms.
Q3. The results showed a mild cortisol response, attributed to weakened HPA axis function and moderate VR stress intensity in obese children, but there was a lack of direct evidence to support it.
Q4. The confounding factors that might affect the results, such as the participants' sleep quality, daily physical activity levels, and family dietary environment, were not adequately controlled.
Q5. The results mentioned that there was no significant difference in HRV between gender and age group, but the interaction effect between the two and its impact on eating behavior were not analyzed.
Q6. It is suggested to further explain the association mechanism between "parasympathetic response" and "sympathetic activation" and eating behavior in combination with specific physiological pathways.
Author Response
Response to Reviewer Comments
We sincerely thank the reviewer for their thoughtful and constructive feedback. Below, we address each point in detail and have made corresponding revisions in the manuscript where appropriate. Changes are marked in the revised manuscript.
Reviewer Comment Q1:
The sample size of this project is relatively small, which may limit the statistical power and extrapolation of the results. It is suggested to supplement the explanation of the basis for estimating the sample size and clearly point out the possible limitations brought by a small sample in the discussion.
Response:
We agree with the reviewer. We have now included a statement in the Methods (Statistical Analysis) section clarifying that this was an exploratory pilot study, and thus, no a priori sample size calculation was performed. Additionally, we have explicitly acknowledged the limitations of our small sample size in the Discussion and Conclusion sections, noting the reduced statistical power and the need for validation in larger cohorts.
Statistical analysis - Changes made on page 6,
This study was designed as an exploratory pilot investigation. The final sample included 30 participants, which is within the range recommended for pilot studies and consistent with prior exploratory research involving heart rate variability and salivary biomarkers)
Discussions - Changes made on page 16
Limits
This study should be interpreted as a pilot investigation, given the relatively small sample size in relation to the complexity of the physiological and behavioral data collected. Although the multimodal design allowed for detailed characterization of autonomic and endocrine reactivity, as well as subjective and behavioral outcomes, the sample size limits the statistical power and generalizability of the findings. The use of exploratory factor analysis and correlation testing was intended to identify preliminary patterns of physiological stress reactivity and their associations with eating behavior traits, rather than to confirm specific hypotheses. Additionally, the absence of a control group and the cross-sectional design preclude causal inference. Future studies with larger, representative samples and longitudinal follow-up are warranted to validate the physiological response profiles observed here and to further investigate their behavioral and clinical significance in pediatric obesity.
Conclusion- Changes made on page 16
Given the exploratory nature and modest sample size of this study, the findings should be viewed as preliminary evidence that requires validation in larger, more representative studies.
Reviewer Comment Q2:
Although VR games have been mentioned as capable of inducing social emotional stress, there has been no systematic comparison of the effects of this model with those of traditional stress paradigms.
Response:
We have expanded the Discussion (VR-based stress model) section to include a brief comparison between traditional stress paradigms (e.g., the Trier Social Stress Test) and VR-induced stressors.
Revisions made on page 13
Several studies have demonstrated that the VR-TSST effectively elicits subjective stress and autonomic responses comparable to those observed in the traditional TSST [30-34]. However, VR-TSST protocols tend to elicit blunted HPA-axis activation, as evi-denced by attenuated salivary cortisol responses in comparison to the traditional TSST [32,33,35]. These findings suggest that while immersive VR scenarios succeed in trigger-ing real-time stress appraisal and sympathetic nervous system reactivity, they may lack the full social-evaluative threat required to robustly stimulate cortisol secretion, possibly due to reduced perceived presence or emotional salience in virtual settings.
A recent meta-analysis comparing traditional and VR-TSST protocols concluded that while subjective and cardiovascular stress responses were largely equivalent, cortisol re-activity was consistently lower in VR-based settings, emphasizing the nuanced dissocia-tion between autonomic and neuroendocrine stress pathways [31]. Repeated exposure to both real and VR-TSSTs further supports this dissociation, showing faster habituation in cortisol responses in virtual conditions [33]. The refined version of VR-TSST, featuring improved immersion, interactive audiences, and realistic social dynamics, showed more robust endocrine responses [34].
Reviewer Comment Q3:
The results showed a mild cortisol response, attributed to weakened HPA axis function and moderate VR stress intensity in obese children, but there was a lack of direct evidence to support it.
Response:
We have now included additional references in the Discussion that support the observation of blunted HPA axis activity in children with obesity/overweight.
Added to page 13,
The blunted cortisol response observed in our study aligns with existing evidence in-dicating hypothalamic–pituitary–adrenal (HPA) axis dysregulation in individuals with obesity. Research has shown that obesity, particularly with central fat accumulation, is associated with attenuated cortisol responses to stress. For example, Therrien et al. [41] demonstrated that both obese and weight-reduced adults exhibited a significantly reduced cortisol awakening response (CAR) compared to lean individuals, with gender and vis-ceral fat distribution being key moderating factors. Similarly, Hillman et al. [42] reported diminished HPA axis reactivity among adolescent females with obesity, suggesting that early-onset obesity may impair neuroendocrine stress responsiveness during critical de-velopmental periods. These findings support the hypothesis that chronic metabolic and inflammatory alterations in obesity contribute to a downregulation of cortisol output, po-tentially as an adaptive or maladaptive response to repeated stressor exposure.. Similarly, Herhaus et al. [43] found that individuals with obesity showed lower cortisol reactivity to psychosocial stress, which was associated with increased food intake, suggesting a dysregulated endocrine response that may facilitate non-homeostatic or emotionally driven eating behaviors. Supporting these findings in younger populations, our previous research [39] using a digital adaptation of the Trier Social Stress Test demonstrated atten-uated cortisol responses in overweight adolescents, along with significant associations between cortisol reactivity and stress-related eating patterns.
Reviewer Comment Q4:
The confounding factors that might affect the results, such as the participants' sleep quality, daily physical activity levels, and family dietary environment, were not adequately controlled.
Response:
We appreciate this observation. In the Limitations section, we have now acknowledged that variables such as sleep quality, habitual physical activity, and home food environment were not controlled in our analysis, which may have influenced stress reactivity and eating behaviors. We suggest these be included in future longitudinal and experimental studies.
Added to page 17,
Moreover, several potential confounding factors that may influence stress reactivity and eating behavior were not adequately controlled. These include participants’ sleep quality, habitual physical activity, and the broader psychosocial and dietary environment within the family context. Such variables could impact both autonomic and HPA axis functioning, as well as eating behaviors, and their omission represents a limitation to the internal validity of the study. Future research should address these confounding factors and include larger, more diverse samples, ideally with longitudinal follow-up, to further to further clarify the physiological and behavioral mechanisms underlying stress reactivity in pediatric obesity.
Reviewer Comment Q5:
The results mentioned that there was no significant difference in HRV between gender and age group, but the interaction effect between the two and its impact on eating behavior were not analyzed.
Response:
Thank you for this suggestion. Given the limited sample size, we did not conduct interaction effect analysis due to low statistical power. However, we now mention this in the Discussion as a direction for future research, and we recommend a larger sample be used to explore age-by-gender interactions on stress reactivity and eating behavior.
Added to page 17,
Future research should aim to address these limitations by recruiting larger samples and incorporating control groups. In future studies, we recommend to explore age-by-gender interactions in stress reactivity and eating behavior, as developmental and hormonal factors may modulate these responses in important ways.
Reviewer Comment Q6:
It is suggested to further explain the association mechanism between "parasympathetic response" and "sympathetic activation" and eating behavior in combination with specific physiological pathways.
Response:
We have enriched the Discussion (Correlations Between Physiological Response Factor Scores and Eating Pattern Measures) section with a more detailed physiological rationale. Drawing on the neurovisceral integration model and polyvagal theory, we discuss how parasympathetic reactivity may enhance interoceptive sensitivity and contribute to cue-driven eating in individuals with low cognitive control. Similarly, we explain that sympathetic activation may
Added to page 16,
Acording to the polyvagal theory[46],, higher vagal tone not only supports social en-gagement and emotion regulation but may also amplify awareness of internal states. In individuals with low cognitive restraint or impaired executive functioning (common in pediatric obesity), this heightened interoceptive sensitivity may contribute to disinhibited or compulsive eating in response to stress or food-related cues.
Added to page 16,
Moreover, studies investigating the relationship between restrained eating, attention to food cues, and autonomic regulation found that restrained eaters show greater sympa-thetic dominance, which may reflect dysregulation in self-regulatory systems. For exam-ple, Meule et al. demonstrated that restrained eating is associated with faster reactions to high-calorie food cues and altered cardiac autonomic balance, suggesting impaired top-down inhibitory control in response to tempting stimuli [49] Similarly, Wu et al. found that lower high-frequency HRV (a marker of reduced parasympathetic tone) was signifi-cantly associated with higher food cravings in adolescents, particularly in those with dif-ficulties in controlling food intake—further supporting the link between autonomic dysregulation and eating vulnerability[50]. From a neurobiological perspective, stress-induced sympathetic activation may promote both restrictive and disinhibited eat-ing patterns, depending on the individual's cognitive control and emotional regulation capacity. Some children may respond to internal arousal and perceived threat by increas-ing cognitive restraint, using top-down control strategies to suppress food intake[51,52]. However, restrained eating is often ineffective in the long term; such individuals are prone to weight cycling and increased BMI, as cognitive control tends to break down in high-stress or food-rich environments, leading to episodes of disinhibited eating.
Added to page 17,
In conclusion, our findings highlight the heterogeneity of physiological regulation in children with obesity. While parasympathetic reactivity may reflect both adaptive (reduced anxiety) and maladaptive (increased UE) processes, sympathetic activation may support compensatory restraint in the face of heightened emotional arousal. Our results extend the application of biopsychophysiological models of eating behavior into immersive stress-induction contexts, such as VR-based paradigms, and suggest that these distinct autonomic profiles may require differential intervention strategies in pediatric obesity management.
We hope that these revisions address all of the reviewer’s concerns and enhance the clarity, rigor, and scientific value of our manuscript. We are grateful for your time and expertise.
Kind regards,
Veronica Mocanu
(On behalf of all authors)

Reviewer 3 Report
Comments and Suggestions for Authors
This report is based on a sound research idea. Unfortunately there are some shortcomings and a number of possible improvements:
Title: the term VR-Induces STRESS is rather strong given the rather mild reactions, in the abstract the term REACTIVITY sounds more appropriate
Introduction: l 111 "few studies have ..." no references
Methods: Age should not be reported with mean and SD, but median(Q1/Q3);
how was inclusion range 8-17 years pre-defined?
VR game: how would a geometric shape be selected, how was a failure vizualized? It may make a big difference it is just a missed opportunity or a "death"-like scene
Questionnaires: why no pre-post comparison for stress level and appetite??
Timing: experiment between 11:00 and 14:00 includes possibly typical lunch time, appetite/hunger etc may very much depend on time of day? This should be tested/vizualized
Statistics: was the EFA based on rank correlations? Was really a factor analysis done (optimizing for independence of latent factors) or PCA (optimizing for variance explained)? Wording is a bit confusing.
Results: As meanRR and HR are inversely related, they do not provide independent evidence
Why are no baseline (pre-VR) saliva results reported?
Figure do contradict table 2, the strong overlap of 95% CIs would exclude a significant effect. Is the outcome presented as median and 95% CI? How computed (e.g. bootstrapping? Additional figures with spread of individual CHANGES may clarify...
l 353 proportion of participants above median was 40%: expected is 50%, so the majority reports 'none' / 1, what is the rational of reporting this?
Table 4 should report complete loadings for both components, scree plot is not adding much, a biplot may be better
Thresholds for normoreactive appear abitrary?
Overall the participants did not on average experience stress, so longer exposure or more challenging VR activities may be needed to elicit more pronounced responses.
Author Response
Response to Reviewer 3
We thank the reviewer for their thoughtful and constructive feedback. We have carefully considered each point and revised the manuscript accordingly to improve clarity, rigor, and transparency. Our responses are outlined below.
- Title Terminology
Comment: The term “VR-Induced Stress” may be too strong given the mild physiological responses observed. “Reactivity” may be more appropriate.
Response:
We appreciate the reviewer’s insightful comment. In response, we have revised the manuscript to better reflect the intensity and nature of the stimulus used in our experimental paradigm. Specifically, we have changed the title of the manuscript to:
“Autonomic and Neuroendocrine Reactivity to VR Game Exposure in Children and Adolescents with Obesity”
to emphasize the focus on physiological and neuroendocrine responses rather than implying a strong stress induction.
Additionally, throughout the manuscript, we replaced the phrase “VR-induced stress” with more appropriate and neutral terms such as “VR game-based stimulation” and “VR game-based experimental model” to describe the immersive virtual reality task. These changes more accurately characterize the experimental conditions as a mild, non-threatening stimulus designed to elicit autonomic reactivity, rather than a classical psychosocial stressor.
We have also clarified this in the Introduction and Methods sections to avoid any potential overstatement regarding the level of stress elicited and to reflect the exploratory, non-threatening nature of the stimulation.
In the present study, we employed a VR game-based experimental model designed to act as a mild, non-threatening stimulation intended to elicit physiological reactivity, rather than a classical psychosocial stressor such as the Trier Social Stress Test (TSST). This paradigm aimed to assess autonomic and neuroendocrine responsiveness without inducing high emotional distress.
The VR game exposure was selected as a low-intensity, emotionally neutral task to mildly activate the autonomic nervous system. This was intentional for ethical considera-tions in pediatric participants. Unlike traditional psychosocial stressors, this protocol was not designed to provoke strong emotional or evaluative stress, but rather to serve as a con-trolled, immersive stimulus for evaluating physiological reactivity in children and ado-lescents with obesity.
- Introduction – Line 111: “Few studies...” lacks references
Response:
Thank you for noting this. We have now added appropriate references to support this claim in the revised Introduction section.
- Methods – Age reporting
Comment: Age should be reported as median (Q1/Q3), not mean ± SD.
Response:
We appreciate this suggestion and have updated age reporting to median and interquartile range as follows:
The median age was 10 years, with an interquartile range (IQR): 9–14 years in methods and results sections.
- Methods – Age range 8–17: how was this defined?
Response:
We have added an explanation to the Methods section: The age range was selected to ensure participants had sufficient cognitive and emotional maturity to understand and engage meaningfully with the virtual reality (VR) game-based experimental task.
- VR Game – Clarify feedback and scene content
Comment: Clarify whether failure was visualized as “death”-like or simply a missed target.
Response:
Thank you for pointing this out. We now clarify in the Methods that the VR game used a non-threatening format; failure to hit the correct shape was indicated by a subtle sound and absence of a reward sound, without any negative or “death-like” visual effects, to ensure suitability for pediatric participants.
- Questionnaires – Why no pre-post comparisons for stress/appetite?
Response:
We appreciate the reviewer’s thoughtful question. In our study design, we intentionally distinguished between chronic and acute psychological states to better understand their relationship to physiological stress reactivity in children with obesity.
We added to methods
To assess different dimensions of psychological stress and eating behavior, we employed a combination of trait-level and state-level questionnaires. Prior to the VR exposure, participants completed the Perceived Stress Scale (PSS), which captures chronic perceived stress over the previous month, and the Three-Factor Eating Questionnaire (TFEQ), which measures trait-like dimensions of eating behavior: cognitive restraint (CR), uncontrolled eating (UE), and emotional eating (EE). Following the VR stimulation, participants completed Visual Analog Scales (VAS) designed to capture acute, momentary states of stress, anxiety, appetite, and craving. This approach enabled us to evaluate both stable behavioral tendencies and immediate subjective responses to the experimental paradigm
We added to Discussions
The use of both trait-level (PSS, TFEQ) and state-level (VAS) measures allowed us to distinguish baseline psychological tendencies from acute subjective responses to VR game exposure. This separation is particularly useful in pediatric populations, where subjective emotional regulation and eating behavior may be context-sensitive and trait-state interactions are relevant for clinical understanding.
- Timing of the experiment (11:00–14:00): potential effect on appetite
Response:
We thank the reviewer for this insightful comment. We have added in the Methods section.
The participants were recruited during day hospitalization visits at Center for Diagnosis, Counseling, and Monitoring of Obese Children at the Emergency Clinical Hospital for Children Sfanta Maria Iasi. All experimental sessions were conducted between 11:00 and 14:00, a period selected to minimize the influence of early morning cortisol peaks while ensuring participant availability during school hours. To reduce variability in appetite and metabolic parameters, a standardized 2-hour fasting period was required prior to testing.
- Statistics – Clarify EFA method and whether PCA was used
Comment: Clarify if exploratory factor analysis (EFA) used PCA or true factor analysis; also whether it was based on rank correlations.
Response:
Thank you for this observation. We clarify that we used Principal Component Analysis (PCA) with oblimin rotation for factor extraction in SPSS, as the goal was to reduce dimensionality and explore physiological response profiles. We have corrected the terminology throughout the manuscript to reflect this accurately.
We added to statistical analyses:
Exploratory factor analysis (EFA) was conducted using principal component extraction (PCA) with oblimin (oblique) rotation, as our aim was to reduce data dimensionality and identify potentially correlated latent components underlying autonomic and endocrine reactivity (%Δ physiological variables). Sampling adequacy was assessed using the Kai-ser–Meyer–Olkin (KMO) test, which yielded a value of 0.671, and Bartlett’s test of spheric-ity which was significant (χ² = 61.9, df = 15, p < 0.001), confirmed the suitability of the data for factor analysis. Based on eigenvalues >1 and the scree plot, three factors were retained. Regression-based factor scores (Regr scores) were then computed in SPSS for each partici-pant to quantify individual expression of the identified factors, which were used in sub-sequent correlational analyses.
- Results – Mean RR and HR are inversely related
Response:
Thank you for your comment. We acknowledge that mean RR interval and HR are mathematically inverse. In our analysis, we prioritized mean RR interval as a standard time-domain measure of HRV that provides greater sensitivity to vagal tone variations. RR is widely used in psychophysiological and stress research due to its relevance for autonomic function interpretation. However, we included HR values where clinically relevant to enhance interpretability and comparison with previous literature.
We added to the Results
Although mean RR interval and heart rate (HR) are mathematically inverse, RR was selected as the primary variable to reflect cardiac chronotropic activity and vagal modulation. RR intervals provide more precise beat-to-beat information and are commonly used in HRV research to assess autonomic responses
- No baseline saliva results shown
Response:
Thank you for this observation. Salivary biomarkers (cortisol and alpha-amylase) were collected at three time points: baseline (before VR exposure), T+10 minutes (post-stimulation), and T+35 minutes (peak time).
We added to Methods.
To capture the endocrine stress response more accurately, we focused on the percent change (%Δ) from post-relaxation (T+10 min) to peak response (T+35 min), as this interval reflects the expected latency of HPA and SAM axis responses to acute stimulation. This approach reduces inter-individual baseline variability and has been used in similar psy-choneuroendocrinological studies [28].
- Figures vs. Table 2: CI overlap and presentation of effects
Response:
Response: Thank you for the observation. In our figures, overlapping confidence intervals (CIs) are present in some comparisons due to sample size and variability. However, the statistical significance of changes between time points was evaluated using the Wilcoxon signed-rank test, which provides more precise evidence of effect. Therefore, while some CI overlap appears visually, the statistical test results reported in Table 2 confirm meaningful differences where present.
- Line 353 – Reporting “40% above median”
Comment: If 40% report scores above median, it implies skew—why highlight this?
Response:
Thank you for the insight. We agree this is potentially misleading. We have revised this section
Perceived Stress Response
Perceived Stress Scale (PSS). Perceived stress was assessed using the Perceived Stress Scale (PSS), and baseline scores prior to VR exposure are presented in Table 2. Elevated stress levels, defined as scores above the 75th percentile, were observed in 8 participants (26.6%)
Stress and Anxiety Assessment (VAS Method). Participants rated their current feelings of stress and anxiety using two Visual Analogue Scale (VAS) questions ranging from 1 (“Not at all”) to 10 (“Extremely”). Due to the non-normal distribution of VAS scores, medians and interquartile ranges (IQR) are reported (see Table 2).VAS scores for stress and anxiety ratings showed a left-skewed distribution, with a subset of participants reporting high stress ratings (VAS Stress > 70, n = 1, 3.3%) and high anxiety ratings (VAS Anxiety > 70, n = 2, 6.7%).
Eating Behavior Response
Appetite and Craving Ratings (VAS Method). Table 2 presents the median and inter-quartile range (IQR: Q1–Q3) for responses to appetite-related VAS items administered after the VR game. Due to the non-normal distribution of VAS scores, medians and interquartile ranges (IQR) are reported (see Table 2). VAS scores for appetite and craving ratings showed a left-skewed distribution, with a subset of participants reporting high hunger ratings (VAS Appetite > 70, n = 7, 23.3%) and high craving ratings (VAS Craving > 70, n = 2, 6.7%)..
- Table 4 – Full loadings and visualization
Comment: Full loadings for both factors should be shown; biplot preferred over scree plot.
Response:
We appreciate the reviewer’s suggestion. In response, we have included the full factor loading matrix (including coefficients below 0.4) in the Supplementary Material_S1, while maintaining the simplified loading table (≥ 0.4) in the main manuscript for clarity. We added a Figure 4 (biplot).
- Thresholds for “normoreactive” classification appear arbitrary
Response:
Conventional thresholds like 10% or 20% aim to define a change that is considered meaningful in the context of a stress response. They represent a noticeable shift from baseline that researchers hypothesize is indicative of physiological activation. The references were added in methods and discussions.
- Mild stress response: was VR stimulation sufficient?
Response:
We agree that the VR stressor was moderate in intensity. This was intentional for ethical considerations in pediatric participants. We have clarified this in the Methods sections:
The VR game exposure was selected as a low-intensity, emotionally neutral task to mildly activate the autonomic nervous system. This was intentional for ethical considera-tions in pediatric participants.
We sincerely thank the reviewer for these detailed insights, which have substantially improved the clarity and depth of our manuscript.
Kind regards,
Veronica Mocanu
(On behalf of all authors)

Round 2
Reviewer 2 Report
Comments and Suggestions for Authors
No addional comments.
Author Response
We sincerely thank the reviewer for their time and previous feedback. We appreciate that no additional comments were raised at this stage.
Reviewer 3 Report
Comments and Suggestions for Authors
Most concerns were met sufficiently, a few still open requests:
- Figures vs. Table 2: CI overlap and presentation of effects
Response:
Response: Thank you for the observation. In our figures, overlapping confidence intervals (CIs) are present in some comparisons due to sample size and variability. However, the statistical significance of changes between time points was evaluated using the Wilcoxon signed-rank test, which provides more precise evidence of effect. Therefore, while some CI overlap appears visually, the statistical test results reported in Table 2 confirm meaningful differences where present.
Please provide figures with the change scores with median and CI!
As for the baseline saliva: Results should be provided to put pre-post changes into perspective.
As for pre-post questionnaires:
We added to Discussions
The use of both trait-level (PSS, TFEQ) and state-level (VAS) measures allowed us to distinguish baseline psychological tendencies from acute subjective responses to VR game exposure. This separation is particularly useful in pediatric populations, where subjective emotional regulation and eating behavior may be context-sensitive and trait-state interactions are relevant for clinical understanding.
To evaluate the STATE part, a reference would be helpful, so I still see this as a limitation of the study design.
Author Response
Response to Reviewer 3
We thank the reviewer for their thoughtful and constructive feedback. We have carefully considered each point and revised the manuscript accordingly to improve clarity, rigor, and transparency. Our responses are outlined below.
- Please provide figures with the change scores with median and CI
Response:
We have added a Figure.
Figure 4. The boxplots illustrate the median and 95% confidence intervals (CI) of the percentage change (%Δ) in heart rate variability (HRV) parameters and salivary stress markers following VR game-based exposure: %Δ RR [A], %Δ SDNN [B], %Δ RMSSD [C], %Δ LF/HF Ratio [D], and salivary markers of stress cortisol, %Δ sCortisol [E] and alpha amylase, %Δ sAA [F].
- As for the baseline saliva: Results should be provided to put pre-post changes into perspective.
Response:
Thank you for noting this. We have corrected the graphs for salivary cortisol and alpha amylase.
Figure 3. Effects of VR game exposure on heart rate variability (HRV): Mean RR [A], SDNN [B], RMSSD [C], and LF/HF Ratio [D], and salivary markers of stress cortisol [E] and alpha amylase [F].
- As for pre-post questionnaires:
To evaluate the STATE part, a reference would be helpful, so I still see this as a limitation of the study design.
Response:
We thank the reviewer for this thoughtful observation. We agree that To evaluate the STATE part, a reference would be helpful, so I still see this as a limitation of the study design. We have now added relevant references to the Discussion section to better support the design rationale and the limition.
We added to Discussions:
The use of both trait-level (PSS, TFEQ) and state-level (VAS) measures allowed us to differentiate between stable psychological tendencies and acute subjective responses fol-lowing VR game exposure. This separation is especially important in pediatric popula-tions, where contextual sensitivity and trait–state interactions influence emotional regula-tion and eating behavior. This differentiation is supported in the literature. Prior research-es emphasize the importance of using real-time or momentary assessment tools, such as VAS, to evaluate state responses, especially in studies on emotional regulation and eating behavior. [34-36].
We added to Limitations
Furthermore, while we employed both trait-level (e.g., PSS and TFEQ) and state-level (e.g., post-exposure VAS ratings) psychological measures, the absence of validated, pre-post state-specific instruments for all subjective variables (such as appetite and stress) limits our ability to comprehensively evaluate short-term emotional reactivity. Although VAS tools are commonly used to capture acute responses in pediatric settings, future studies should consider including standardized, validated state measures to better differ-entiate transient versus enduring psychological patterns.
We sincerely thank the reviewer for these detailed insights, which have substantially improved the clarity and depth of our manuscript.
Kind regards,
Veronica Mocanu
(On behalf of all authors)
